# Unlocking hidden biomolecular conformational landscapes in diffusion models at inference time

**Daniel D. Richman**\*
Stanford University
ddrichma@stanford.edu

**Jessica Karaguesian**\*
Stanford University
jkara@stanford.edu

**Carl-Mikael Suomivuori**†
Stanford University
carl-mikael.suomivuori@yale.edu

**Ron O. Dror**
Stanford University
rondror@cs.stanford.edu

## Abstract

The function of biomolecules such as proteins depends on their ability to interconvert between a wide range of structures or "conformations." Researchers have endeavored for decades to develop computational methods to predict the distribution of conformations, which is far harder to determine experimentally than a static folded structure. We present ConforMix, an inference-time algorithm that enhances sampling of conformational distributions using a combination of classifier guidance, filtering, and free energy estimation. Our approach upgrades diffusion models—whether trained for static structure prediction or conformational generation—to enable more efficient discovery of conformational variability without requiring prior knowledge of major degrees of freedom. ConforMix is orthogonal to improvements in model pretraining and would benefit even a hypothetical model that perfectly reproduced the Boltzmann distribution. Remarkably, when applied to a diffusion model trained for static structure prediction, ConforMix captures structural changes including domain motion, cryptic pocket flexibility, and transporter cycling, while avoiding unphysical states. Case studies of biologically critical proteins demonstrate the scalability, accuracy, and utility of this method.

## 1 Introduction

Biology depends on molecular flexibility. Proteins, RNA, DNA, and other components of biological systems adopt dynamic 3D conformations, which are key to function [7, 8]. Understanding and modeling these dynamics informs both basic biology research and applications to medicine, agriculture, and other areas, and is therefore a major aim of both experimental and computational scientists. The statistical distribution of the atomic configurations of a molecular system in equilibrium at a given temperature is determined by their potential energies. Specifically, a state with lower potential energy is exponentially more probable than a higher potential energy state, as determined by the Boltzmann distribution [31]. To sample this distribution, scientists have invested decades of effort in creating methods such as Monte Carlo samplers and molecular dynamics simulations (MD), but these methods are typically extremely slow and suffer from accuracy problems [25, 31, 32].

More recently, machine learning models have revolutionized biomolecular structure prediction, and efforts are underway to extend these advances to train models that do not merely predict static structures but generate diverse conformations from the true Boltzmann distribution [1, 11, 12, 17,

---

\*Equal contribution.
†Present affiliation: Yale School of Medicine.

39th Conference on Neural Information Processing Systems (NeurIPS 2025).

35]. As the field evolves, new questions arise about how best to take advantage of these models, access the information they contain, and target opportunities for further improvement. We ask the following question: *Given a pretrained generative structure model, what strategy should be deployed at inference time to efficiently and accurately sample its conformational landscape?*

In this work, we present an enhanced sampling algorithm, ConforMix, that can be applied to diffusion-based biomolecular structure prediction such as the AlphaFold 3 family. ConforMix reveals hidden conformations and estimates free energies more efficiently than the default sampling from diffusion models. Our approach is based on the hypothesis that the probability distribution of generative models contains more information than is efficiently sampled. Importantly, even a generative model that *perfectly* sampled the true Boltzmann distribution—a major long-term goal in machine learning for biology—would, for many uses, still require enhanced sampling algorithms to fruitfully explore it. For instance, correct sampling of the tight-binding proteins barnase and barstar at standard state would generate an unbound structure with probability approximately $10^{-14}$ [33]. Directly drawing samples from the distribution would be an extremely inefficient way to estimate this probability, determine transition states along the binding pathway, or understand the effects of environmental factors in modulating this affinity [30, 39].

**Contributions** We introduce, implement, and benchmark ConforMix. Our contributions include:

- A novel algorithm combining twisted sequential Monte Carlo, which performs asymptotically exact sampling of conditional distributions, with an automated procedure for exploring the diffusion landscape, using conditional sampling as a subroutine. Optionally, a statistically optimal sample reweighting algorithm, applied to diffusion models for the first time, can be used to reconstruct the unconditional distribution from the conditional samples.

- Dramatically improved inference-time sampling in structure prediction models. We implement ConforMix in Boltz-1, an AlphaFold 3–like diffusion model that typically predicts only one distinct conformation for a given input. ConforMix-Boltz generates realistic and diverse conformations for a variety of proteins, without prior knowledge of important degrees of freedom. In prior work, conditional sampling on biomolecular diffusion models has required additional input information, such as experimentally measured pairwise distances.

- Efficient free energy estimation. We also implement ConforMix in BioEmu, a diffusion-based model for conformational sampling, and compare free energy estimates from BioEmu with and without ConforMix. Using ConforMix boosts the speed of free energy estimation.

## 2 Related Work

Computational methods have been used to sample molecular conformations for decades. The well-known Metropolis-Hastings algorithm [23] for Monte Carlo sampling was first used to simulate the statistical mechanics of particles in a box. Elastic network models represent a protein as low-resolution beads connected by springs and recover the principal motions that would occur in this representation, which often are representative of motions of the actual system [20]. More sophisticated methods, such as coarse-grained or all-atom molecular dynamics simulation, model the physical interactions with progressively more detail and have been used to generate detailed probabilistic distributions for a number of proteins [18, 24].

More recently, machine learning models such as AlphaFold 3 [1] have revolutionized the problem of static protein structure prediction. AlphaFold 3 is the most prominent example of a class of diffusion-based structure prediction methods that excel at generating the single most likely protein structure for a given amino acid sequence. While AlphaFold 3 generates five samples per run, for well-ordered proteins these samples are often almost identical and are better understood as very similar predictions of the same static structure rather than a representation of conformational variety.

Using such models to model the *distribution* of conformations, however, is a task still in its infancy. The most well-studied approaches involve subsampling or otherwise perturbing the multiple sequence alignment (MSA) input [13, 15, 27, 35] or templates input [2, 4, 9] to AlphaFold and related models. These inputs specify, respectively, sequence co-evolutionary information and related 3D structures. Emphasizing different sequence or structural features in the input has empirically produced significant conformational diversity. Subsampling methods, however, suffer from fundamental limitations. First, subsampling necessarily involves reducing the information available to the model, which often

produces poor-quality outputs. Second, molecular conformations are continuous, while subsampling of sequences or templates is a discrete operation. Third, the structure distribution accessed by subsampling methods is not well defined, and despite years of study there is no clear means of reconstructing actual probabilistic ensembles.

Beyond input-perturbation methods, other efforts have involved building new models to predict conformations [6, 11, 17]. Strategies include leveraging molecular dynamics simulations or experimental data, either during training or at runtime [3, 16, 19, 21, 34]. While these approaches have shown promise, they do not yet accurately reproduce the thermodynamic ensembles of arbitrary proteins.

Based on the successes of MSA subsampling and template perturbation methods, we hypothesized that diffusion models have learned richer energy landscapes than are generally recovered and that these landscapes can be accessed via a more sophisticated sampling approach. As improvements in model training lead to more accurate learned landscapes, it will become increasingly valuable to not only access the most dominant configurations from the model but also extract rich information about transition mechanisms and probabilities. We aim to open the door to this type of insight.

## 3   Methods

Given a biomolecular structure diffusion model $p(x|s)$, where $s$ is the input biological system provided to the model (for instance, the amino acid sequence of a protein and its multiple sequence alignment) and $x \in \mathbb{R}^{n \times 3}$ specifies the predicted atomic coordinates, our objective is to efficiently explore and characterize the distribution $p$.

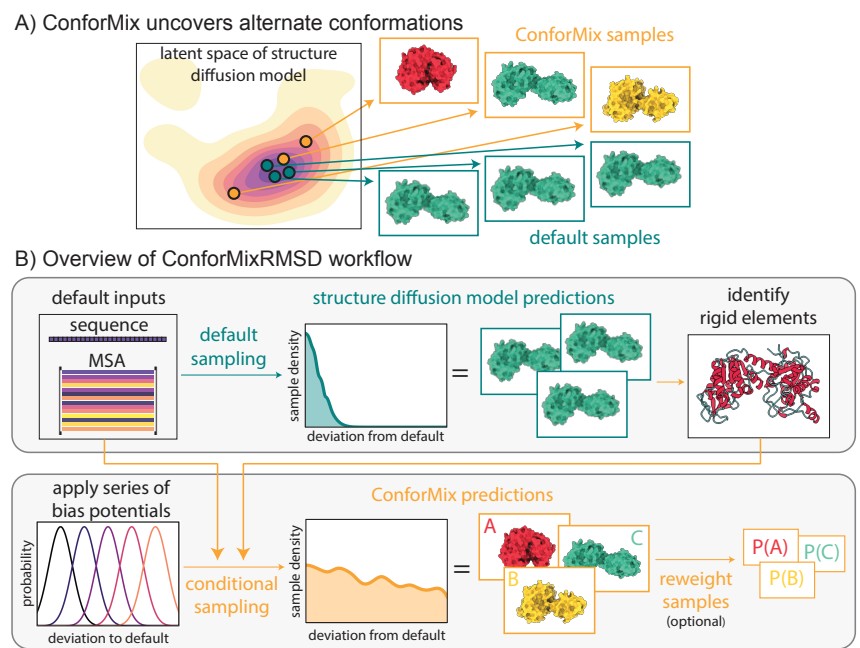

Figure 1: (A) ConforMix adds conditioning to diffusion-based structure prediction models, enabling deeper and more efficient exploration. (B) ConforMix uses a series of bias potentials to target new states. ConforMixRMSD is an instantiation that biases sampling away from default predictions to sample conformational transitions without requiring any user intervention. Sample reweighting can be applied to recover the ground state distribution.

### 3.1   ConforMix

Diffusion models learn probability distributions, and the standard sampling procedures, which we refer to as *default sampling*, are designed to draw i.i.d. samples from this distribution. To explore further and more efficiently than is possible with default sampling, we employ conditional sampling, which is widely used with diffusion models to guide sampling toward particular regions of the learned

probability distribution [5, 10]. As we will see, conditional sampling offers clear benefits even without specialized prior knowledge about the protein of interest. While time-dependent classifier guidance and classifier-free guidance are commonly used in other diffusion contexts, these methods are not well suited to our needs because they require either additional training or special architectural choices, which are impractical for our needs.

Instead, to devise a purely inference-time method, we turn to Twisted Diffusion Sampling [37], a form of sequential Monte Carlo (SMC), to guide conditional generation. SMC has advantages over more typical guidance methods, including (a) requiring only a time-independent guidance function, which enables us to easily describe conditions as a function of the fully denoised state of the system, and (b) providing an asymptotic guarantee of sampling the true target conditional distribution in the limit of many particles. Twisted diffusion sampling works in conjunction with an underlying diffusion model that learns to sample a target data distribution $q(x_0|s)$. Diffusion models define a forward noising process, operating over a sequence of $T$ steps, that constructs an extended distribution $q(x_{T:0}|s) = q(x_T|x_{T_1}) \cdots q(x_1|x_0)q(x_0|s)$. The diffusion model $p_\theta(x_t, t|s)$ can then be trained to approximate the score function $\nabla_x \log q_t(x|s)$, which is used to simulate the noising process in reverse. We emphasize that ConforMix itself does not involve additional model training.

Instead, twisted diffusion operates as a particle filter. The sampling algorithm is instantiated with a series of guidance functions $y_j(x) = \exp(-U_j(x))$, where $U_j$ is a potential. For example, many motions can be described in terms of the distance between two groups of atoms $A$ and $B$, so we might select a potential

$$U_j(x) = \frac{\alpha}{2} \left( ||\vec{\mu}_A - \vec{\mu}_B|| - \lambda_j \right)^2 \tag{1}$$

where $\alpha$ is a strength parameter, $\vec{\mu}_A, \vec{\mu}_B$ are the centroids of the atom groups, and $\lambda_j \in [\lambda_{\min}, \lambda_{\max}]$ is a target distance. The corresponding score function

$$\nabla_x \log y(x) = -\alpha \left( ||\vec{\mu}_A - \vec{\mu}_B|| - \lambda_j \right) \nabla_x ||\vec{\mu}_A - \vec{\mu}_B|| \tag{2}$$

is easy to compute. During inference, at each denoising timestep $t$, we construct an inexpensive estimate of the fully denoised state $\hat{x}_0$, which we plug into $y_j$ to approximate the potential of the current state. We then compute $\nabla_{x_t} y_j = (J_{x_t} \hat{x}_0)^T \nabla_{\hat{x}_0} y_j$ and use this signal to guide $x_t$ towards $p(x|y_j, s)$. By simultaneously denoising several such particles $x_t$ and periodically resampling them to maintain correct sampling, we can obtain approximate samples from the desired conditional distribution. For more details, we refer the reader to Algorithm S1 in the Supplement and to Algorithm 1 in [37], which provides an asymptotic guarantee that the sampling approaches the true conditional distribution as the number of particles becomes large.

In our general framework, ConforMix, twisted diffusion sampling can be used to sample from nearly arbitrary conditional distributions. To select suitable conditioning, we are motivated by two biological use cases: (1) automated exploration of the conformational landscape, where a user provides only an input system (i.e. amino acid sequence, and optionally MSA and/or templates), and (2) user-defined exploration of conformational states, where a user wishes to examine a specific hypothesized motion or state. In the rest of this paper, we focus primarily on the first case, but we note that ConforMix can be customized to scan targeted degrees of freedom.

### 3.2 ConforMixRMSD

Our approach to the automated exploration problem, which we call ConforMixRMSD, is described in Algorithm 1. Here we assume that a user wishes to perform undirected exploration of the protein conformational distribution learned by the underlying model. The idea of ConforMixRMSD is to simply generate the most probable samples that are sufficiently different from the default prediction $x_d$. AlphaFold 3–family generative models often have probability distributions $p(x|s)$ that are highly concentrated around one conformation. After generating an initial conformation $x_d$ with default sampling, we construct a series of guidance potentials that steer the probability distribution away from $x_d$ using a root mean square deviation (RMSD) metric: $U_j(x) = \frac{\alpha}{2} \text{RMSD}(x, x^{ref})^2$, where $\text{RMSD} = \min_{R, \mathbf{t}} \frac{1}{N_{atoms}} \sum_{a=1}^{N_{atoms}} ||x_a - (R\mathbf{x}_a^{ref} + \mathbf{t})||^2$, and $R \in \text{SO}(3)$ and $\mathbf{t} \in \mathbb{R}^3$ respectively rotate and translate $x$ to best align to $x^{ref}$. RMSD is differentiable and computable in closed form via the Kabsch algorithm. The fastest motions of a protein, which are easier to predict are often fluctuations of disordered loop regions. To avoid exclusively sampling these motions, we apply a

rigid-element mask: we only compute RMSD on amino acids that are part of secondary structure elements, $\alpha$ helices or $\beta$ sheets. All amino acids still remain free to move during sampling.

Because the realistic range of motion varies per protein and is not known a priori, we seek to discover the flexibility of each protein by generating structure predictions spanning the range 0 to 20 Å RMSD to $x_d$. After samples are generated, we filter those that are physically implausible. Specifically, we reject samples where any 10-residue sliding window has an average pLDDT value of more than 20% below that of the default prediction, as well as structures with clashes. The sliding window approach detects local non-physical perturbations from sampling beyond the protein's flexibility range; indeed, the amount of filtering increases as we bias to larger RMSD to $x_d$ (Figure S6).

---

**Algorithm 1** ConforMixRMSD for exploration of conformational landscapes

---

**Input:** Biomolecular structure prediction model $p$, input system $s$, target RMSD values $\mathcal{R}$, constraint
   strength $\alpha$, number of samples per RMSD $N_{samples}$
**Output:** Samples $\{x_{R,i}\}$ for all $R$ and $i$
 1: $x_d \leftarrow p(x \mid s)$        # *Predict a structure using default sampling*
 2: $mask \leftarrow \text{RIGIDELEMENTS}(x_d)$     # *Identify atoms within secondary structure elements*
 3: $g_r(y \mid x) \leftarrow \exp\left(-\alpha\left(\text{RMSD}(x, x_d; mask) - r\right)^2\right)$     # *Define conditioning potential*
 4: **for** $r \in \mathcal{R}$ **do**
 5:     **for** $i = 1, \cdots, N_{samples}$ **do**
 6:         $x_{r,i} \leftarrow \text{CONFORMIX}(x; g_r, s)$
 7:     **end for**
 8: **end for**
 9: **return** $\{x_{r,i}\}_{i=1}^{N_{samples}}$ for all $r \in \mathcal{R}$

---

### 3.3   Sample reweighting and free energy estimation

While some scientific problems require only qualitative information about accessible conformational states and flexibility, others require estimates of statistical quantities such as free energy differences. For this reason, we wish to recover the free energy landscape of structure prediction models.

Samples generated from ConforMix are sampled from a series of conditional probability distributions $p(x|y_j, s)$. To form a clear picture of this latent distribution, we wish to reweight our mixture of conditional samples to the unconditional distribution. Combining samples from multiple conditional distributions requires estimating the partition functions $p(y_j)$. An unbiased estimator of this quantity is available directly from twisted diffusion sampling, but in practice these estimates are noisy, particularly for rare samples, and therefore convergence is slow.

To solve the challenge of estimating the partition functions $p(y_j)$, we turn to a statistical tool that to the best of our knowledge has not previously been applied in the context of diffusion models. The multistate Bennett acceptance ratio (MBAR) free energy estimation algorithm combines sample information from multiple conditional distributions to jointly estimate the probabilities $\{p(y_j)\}$, up to a (negligible) constant factor [28, 29]. In practice, this means that given sufficiently accurate samples from a series of overlapping distributions, such as the conditional distributions of ConforMix, we can reconstruct the unbiased model landscape in those areas. We discuss free energy estimation further in Section 4.4. The mathematical details of MBAR are deferred to the appendix.

## 4   Results

We evaluate ConforMix in multiple settings to explore the following questions:

   Q1. How do conformational samples generated via inference-time guidance with fixed inputs
   (sequence, MSA, etc.) compare in quality and coverage to samples generated via methods
   that require altering the input MSA?

   Q2. Can relevant degrees of freedom be extracted from the energy landscape of a model trained
   for single structure prediction?

   Q3. As future generative models improve accuracy in predicting the true Boltzmann distribution,
   can enhanced sampling algorithms like ConforMix provide rapid quantitative information?

## 4.1 ConforMixRMSD recovers experimentally observed conformations

We first implement ConforMix in Boltz-1 [36], an open-source diffusion-based structure prediction model similar to AlphaFold 3. Importantly, Boltz was trained on the Protein Data Bank (PDB), and not with any other dynamics information. While the default Boltz sampling can generate multiple samples, they are generally highly concentrated and are more usefully thought of as multiple approximations of the same static structure than as samples from the true conformational distribution of the protein. To address Q1, we collect samples using ConforMixRMSD with Boltz (ConforMixRMSD-Boltz) as the underlying structure prediction model.

We examine performance of ConforMixRMSD-Boltz on proteins that exhibit different types of conformational changes: (a) domain motion, (b) transporter cycling, (c) cryptic pocket formation, and (d) fold switching. The set of 38 proteins that exhibit domain motions are the combination of proteins curated by Lewis et al. [17] and in OC23 by Kalakoti et al. [13]. The set of 15 membrane transporters that exhibit *inward-open* and *outward-open* conformations were curated in TP16 by Xie & Huang [38]. The set of 31 proteins that exhibit cryptic pocket formation were curated by Lewis et al. [17]. The set of 15 fold switchers proteins are a subset of those curated by Porter et al. [26].

On these sets, Boltz alone typically recovers only a single conformation. Sampling from Boltz with ConforMixRMSD, however, reveals that the hidden energy landscape contains degrees of freedom that transition to alternative conformations seen in experiment and deposited in the PDB (Figure 2). As comparators, we implement three MSA-modification protocols in Boltz: AFCluster [35], AFSample2 [13], and CF-random [15] (see details in Supplement). We also compare to default Boltz sampling with $1,000$ samples. On the domain motion, transport, and cryptic pocket sets, ConforMixRMSD recovers more alternate conformations than all baselines—as measured by RMSD and TM-scores to PDB structures (Tables 1, S3 and Figures 2, S1, S4, S7-S10)—demonstrating its power explore conformational space in a novel manner. MSA-based methods show stronger performance on fold switchers, suggesting such rearrangements are be better captured discrete input modulation, whereas ConforMixRMSD is better suited to continuous (e.g. domain, transporter, pocket) transitions.

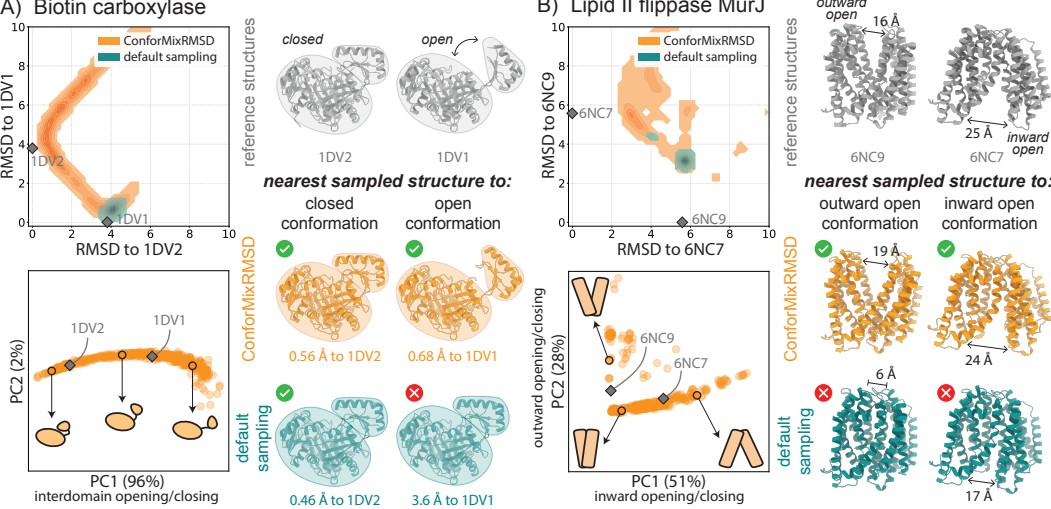

Figure 2: ConforMixRMSD uncovers conformational states that are not sampled by default. Conformational sampling of (A) a domain motion protein and (B) a membrane transporter. For each system, Top Left: density of sampling relative to reference experimental structures. Bottom Left: projection of the ConforMixRMSD sampled structures (orange) onto the first two principal components computed from their internal atomic distances. Experimentaly determined reference structures (grey) are projected onto the same space. Right: reference structures and the closest structure generated by each sampling approach (lowest RMSD).

We also find that recovery of experimentally observed states from ConforMixRMSD-Boltz sampling is competitive with BioEmu [17] (Figure S2), a model trained to generate conformational ensembles,

Table 1: Coverage of experimentally-determined reference conformations by method and dataset

| | | Domain motion (n=38) | Membrane transporters (n=15) | Cryptic pockets (n=31) | Fold switching (n=15) |
|---|---|---|---|---|---|
| **Worst-matched reference conformation** (harder task) | Default Boltz sampling | 0.33 (±0.14) | 0.13 (±0.17) | 0.15 (±0.12) | 0.13 (±0.17) |
| | AFCluster-Boltz | 0.46 (±0.15) | 0.19 (±0.19) | 0.35 (±0.18) | **0.27 (±0.23)** |
| | CF-random-Boltz | 0.51 (±0.17) | 0.20 (±0.20) | 0.39 (±0.16) | 0.20 (±0.20) |
| | AFsample2-Boltz | 0.44 (±0.15) | 0.20 (±0.20) | 0.33 (±0.17) | 0.10 (±0.15) |
| | **ConforMixRMSD-Boltz** | **0.69 (±0.15)** | **0.33 (±0.23)** | **0.45 (±0.18)** | 0.13 (±0.17) |
| **Best-matched reference conformation** (easier task) | Default Boltz Sampling | 0.94 (±0.07) | 0.59 (±0.23) | 0.94 (±0.08) | 0.60 (±0.27) |
| | AFCluster-Boltz | 0.92 (±0.09) | 0.60 (±0.27) | **0.97 (±0.05)** | 0.60 (±0.27) |
| | CF-random-Boltz | 0.87 (±0.12) | 0.60 (±0.23) | **0.97 (±0.05)** | 0.60 (±0.27) |
| | AFsample2-Boltz | 0.90 (±0.08) | 0.67 (±0.23) | 0.93 (±0.08) | **0.70 (±0.30)** |
| | **ConforMixRMSD-Boltz** | **0.97 (±0.04)** | **0.79 (±0.19)** | 0.94 (±0.08) | 0.67 (±0.23) |

Coverage at X% measures the fraction of proteins with samples matching a reference conformation within X% of the RMSD between reference structures. Displayed are values of **coverage evaluated at 50%** of the reference-to-reference RMSD. We evaluate coverage separately for best-matched and alternate (worst-matched) states. Error bars are 95% confidence intervals over 1,000 bootstraps.

although we note that many of the evaluated proteins were held out of training for BioEmu but were likely present in the Boltz training set.

### 4.1.1 Runtime performance

Due to the additional cost of the guidance and resampling steps in Twisted Diffusion Sampling, sampling with ConforMix takes $\sim$ 3x the wall clock time compared to default sampling. However, individual ConforMix samples are much more informative than default samples. ConforMixRMSD can scan large motions of a protein within the space of a few dozen samples, which takes just a few minutes for a moderately sized protein. For many such proteins, default sampling would not reveal such motions even after thousands of samples. Timing statistics are provided in Table S1.

### 4.2 ConforMixRMSD samples conformational transitions of domain motion proteins

It is important for any structural sampling method to distinguish meaningful conformational fluctuations from random perturbations. Consider a protein that exhibits an open-to-closed domain motion. Consider two sampling approaches: (A) generates structures that map the domain transition from open-to-closed—ideally with lower energy wells at observed dominant conformations—while (B) generates structure variation in many directions. While both methods may sample the reference open and closed states of this protein at the same RMSD, approach A is clearly more useful as it identifies the biologically significant motions. Approach B is difficult to use in practice, as the reference conformations are not typically known and thus identifying relevant conformations from the collection of heterogeneous variation is not tractable.

To evaluate whether sampling methods explore biologically relevant conformational landscapes versus generating diffuse structural noise, we analyze the variance of generated ensembles using principal component analysis on pairwise structural distances. We project experimental PDB reference structures onto the principal components to assess whether the major axes of structural variation in the samples correspond to observed conformational transitions. Focusing on the domain motion set, where proteins are known to exhibit a predominant motion between states, we find that ConforMixRMSD tends to produce more organized sampling that highlights the transitions between known conformations, while MSA-based methods typically yield less-interpretable diffuse sampling (Figures 3, S3, S11).

As discussed in the previous section, Boltz sampling generates limited structural diversity, typically clustering near one experimental state. However, we find that the limited variance that does exist between samples often corresponds to the direction of motion needed to reach the alternative conformational state—but lacks the magnitude to actually traverse the transition. This suggests the model recognizes the relevant degrees of freedom but undersamples them. MSA-based approaches present the opposite problem: they generate abundant structural diversity and can more frequently recover both reference structures, but sampling often doesn't map to relevant motion. MSA-based methods show lower values for both the degree of sample variance explained by top principal components and

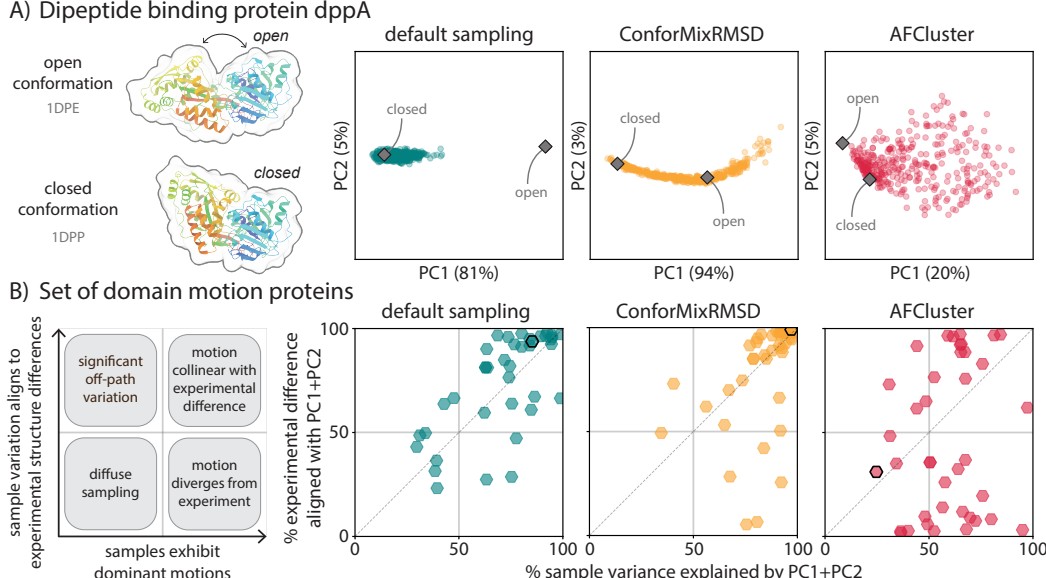

Figure 3: ConforMixRMSD samples domain motion, revealing transitions consistent with experiment while avoiding noisy paths. (A) Principal component analysis of pairwise $C\alpha$ distances of samples generated for a domain motion protein, dppA. Default sampling extends toward but does not reach the open conformation, while ConforMixRMSD traces an opening/closing path. While AFCluster generates open and closed states, its sampling of many other large fluctuations makes it harder to identify the relevant motion. (B) Analysis of variance of samples generated by default sampling, ConforMixRMSD, and AFCluster, all used with Boltz. Each point describes results for one protein. Default sampling and ConforMixRMSD tend to exhibit concentrated variance between samples that align with the direction of domain motion between reference structures. Note that this metric captures alignment of the sampling direction with experimental conformational differences, but not the extent of sampling along that direction——a method may align well without sampling both experimental conformations. AFCluster often produces samples whose structural differences do not match the direction of domain motion between references, or lack dominant directions of variance—indicating off-path sampling. Black-outlined points denote dppA.

the principal component alignment with experimental transitions. These methods generate predictions diffusely across structure space, sometimes matching reference states, but without concentrating sampling along pathways likely to exist between observed states.

ConforMixRMSD, by contrast, often generates substantial conformational diversity while concentrating that diversity along experimentally relevant transitions. For the dipeptide binding protein dppA (Figure 3A), for instance, 96% of the variance in ConforMixRMSD samples is explained by a single principal component that directly corresponds to the domain hinging motion between experimental states. More generally for domain motion proteins (Figure 3B), the top two principal components of each ConforMixRMSD ensemble generally explain both a high fraction of the sample variance (indicating the method generates focused diversity) and a high fraction of the variance between experimental reference structures (indicating this diversity aligns with known conformational changes). When this is the case, conformational ensembles are readily interpretable: their major modes of structural variation correspond to biologically meaningful motions. PCA plots for all proteins in the domain motion set are provided in Figure S11.

As an alternate metric, the "fill ratio" introduced by [13] measures how continuously samples cover the conformational path between reference states based on TM-scores. ConforMixRMSD sampling results in higher fill ratios than the other evaluated methods (Figure S5). We also compute geometric quality metrics on ConforMix-generated samples and observe slightly elevated rates of outliers compared to default sampling, but most samples appear reasonable (Table S2).

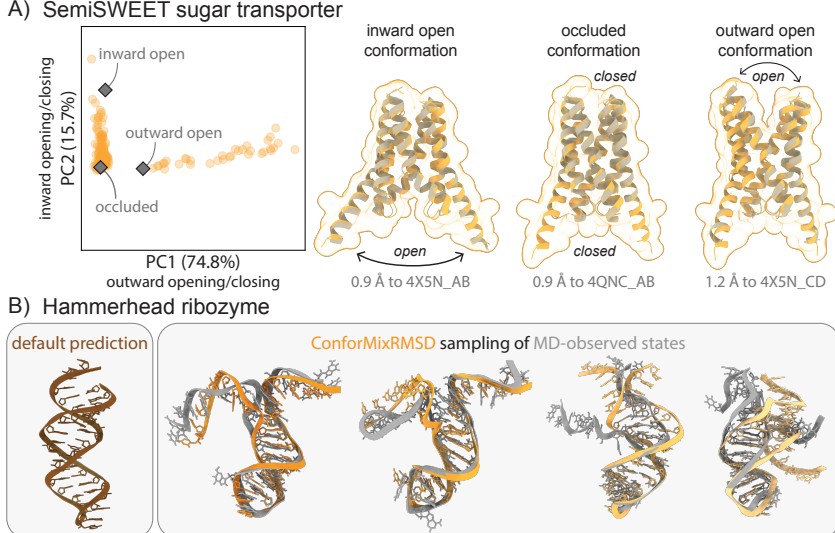

Figure 4: Exploration of biological macromolecules of interest. (A) ConforMixRMSD-Boltz recovers all three experimentally determined conformations of the SemiSWEET transporter. Default Boltz sampling recovers only the occluded state. PCA demonstrates how the samples capture the major motions as the transporter opens to the inward (intracellular) or outward (extracellular) sides. (B) Preliminary application of ConforMixRMSD-Boltz to RNA structure shows it can recapitulate MD-observed transitions.

### 4.3 ConforMixRMSD recovers heterogeneous states in multi-chain and RNA systems

We further evaluate ConforMixRMSD on select biomolecular systems known to exhibit interesting heterogeneity. First, we consider a multi-chain transporter protein, the SemiSWEET sugar transporter, which cycles through three distinct states to transport substrate across a membrane (Figure 4A) [14]. The inward- and outward-open states enable substrate uptake and release, while an occluded state that is closed to both sides allows the substrate to pass through the transporter. Thus far, ML-based conformational sampling approaches have primarily been effective on single-chain proteins. Boltz by default predicts only the occluded state for this system. With ConforMixRMSD sampling, however, Boltz recovers all three states. PCA analysis and visual inspection of sampled structures reveal intermediate states between each known conformation, forming a conformational "trajectory".

While most of our testing focuses on proteins, ConforMix can be applied to any biomolecular system as long as it is supported by the underlying model. We demonstrate sampling of the hammerhead ribozyme (Figure 4B), a small catalytic RNA molecule that adopts multiple conformations. These conformations have not been deposited in the PDB, so instead we take molecular dynamics simulations from [22] as reference data (the grey structures in Figure 4B represent MD frames). ConforMixRMSD-Boltz visually recapitulates key aspects of the unzipping transition observed in the MD simulations. The sampled structures qualitatively capture the conformational trajectory, suggesting that ConforMix can extract meaningful conformational heterogeneity from RNA systems, though comprehensive evaluation across RNA systems would be needed to fully characterize performance.

We conclude that (1) Boltz's learned conformational landscape is rich enough to be of practical utility on many protein systems of interest, and (2) ConforMixRMSD enables inference-time access to this landscape without the disadvantages associated with MSA-modification methods.

### 4.4 Rapid free energy estimation with BioEmu

To illustrate the ability of ConforMix to improve sampling in a variety of situations, we implement it in BioEmu [17] as well. Unlike Boltz, BioEmu produces broad distributions of conformations with default sampling. However, default sampling is an inefficient way to observe rare events or estimate quantities such as free energy differences, so we assess the ability of ConforMix to rapidly estimate free energies from BioEmu.

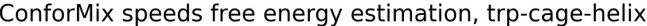

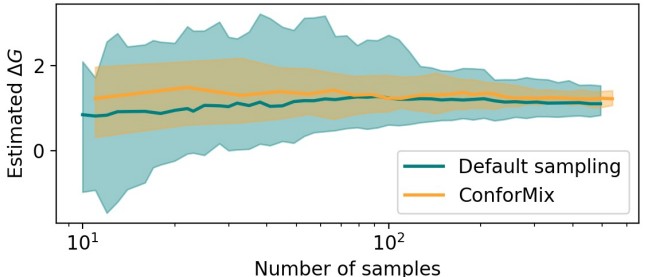

Figure 5: ConforMix sampling in BioEmu enables faster convergence of free energy estimates than default sampling. Using a series of guidance potentials based on RMSD to the native state enables systematic collection of samples that, in turn, enable more rapid free energy estimation. The free energy difference estimated is between the lowest free energy RMSD value (3Å to the reference state) and a more extended conformation at 7.5Å. 90% confidence interval from 50 bootstraps is shown.

While ConforMix can be used to study states that are not seen at all in unbiased sampling, for purposes of comparison we consider a set of transitions that can also be observed without enhanced sampling. Specifically, we evaluate ConforMix on local unfolding proteins from BioEmu, which contain structural motifs that undergo transitions between ordered and disordered states. Because BioEmu does not produce a single consensus structure from default sampling, here we instead use RMSD to the folded reference structure as the basis for defining a series of guidance potentials (see Supplement). We compare the rate of convergence of i.i.d. probabilities from ConforMix estimates with MBAR reweighting to unbiased sampling in BioEmu (Figure 5). Specifically, we estimate the free energy difference between the lowest free energy RMSD value and a specific less folded state. We observe that ConforMix produces estimates close to the true unbiased estimates and converges significantly more quickly.

## 5 Discussion

Predicting biomolecular dynamics represents a major frontier for machine learning. By enhancing inference-time sampling in diffusion models, ConforMix rapidly reveals hidden states and continuous motions without prior knowledge from the user. It also enables characterization of the underlying free energy landscape of the models, which we anticipate will assist in future training and evaluation. The broader impact of tools like this is initially in an improved understanding of basic biology and then in applications to medicine and other areas.

The primary limitation of ConforMix, like other enhanced sampling methods, is that it depends on the robustness and utility of the underlying energy landscape it samples. While Boltz has learned a useful landscape for many proteins, its default sampling typically does not adequately explore the landscape. ConforMix sampling enables us to identify major missing states: known experimental conformations that do not exist with realistic probabilities in the Boltz probability distribution. Other limitations include potential systematic errors due to the inexact sampling of twisted diffusion in the non-asymptotic regime and the statistical uncertainty associated with MBAR. We note that ConforMix has more general uses beyond the specific instantiation of ConforMixRMSD. For instance, users can supply more informative biasing potentials, perhaps based on experimental evidence. If desired, ConforMix can be used in combination with input-modification approaches such as MSA subsampling, although we leave that exploration to future work.

ConforMix is fundamentally flexible. It is not restricted to proteins, and it can handle complexes of multiple molecules. It is able to operate in both all-atom and backbone-only diffusion methods. It is orthogonal to model training. We anticipate that as models mature and energy landscapes approximate the true Boltzmann distribution more closely, inference-time sampling methods such as ConforMix will only become more valuable, both for direct study of protein systems and for model development.

The code for this project is available at github.com/drorlab/conformix.

# 6 Acknowledgments

We thank the anonymous reviewers for their valuable suggestions and feedback on the manuscript. We thank Karl Krauth, EJ Fine, Ayush Pandit, Shawn Costello, Masha Karelina, Brian Trippe, Hunter Nisonoff, and Bowen Jing for scientific discussions.

This work was supported by NIH grant R35GM158122. D.D.R. acknowledges support from the National Science Foundation Graduate Research Fellowship Program. J.K. acknowledges support from the Knight-Hennessy Scholars program and the Natural Sciences and Engineering Research Council of Canada Graduate Research Scholarship.

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
