# OpenReview forum: "Unlocking hidden biomolecular conformational landscapes in diffusion models at inference time"
_NeurIPS.cc/2025/Conference — NeurIPS 2025 spotlight_

### Official Review · Reviewer_61jC · 2025-06-11

**Clarity:** 2
**Significance:** 4
**Originality:** 3
**Rating:** 5
**Confidence:** 4

**Summary:**

The paper presents ConforMix, a method for guiding diffusion-based biomolecular generative models (e.g., Boltz, AlphaFold3) to produce conformational ensembles—rather than single static structures—at inference time, without any model retraining or fine-tuning.

**Questions:**

1.	The 10-residue sliding window for pLDDT (Section 3.2) is an interesting idea. Why was this chosen over, for example, simply averaging pLDDT over all residues?
2.	In Algorithm 1, are samples with specific RMSD thresholds R guaranteed? For example, in Figure 2A, there appear to be no samples within 1Å RMSD for 13PK, despite this threshold being mentioned on line 190.

**Ethical Concerns:**

["NO or VERY MINOR ethics concerns only"]

**Final Justification:**

I maintain my "Accept" score. The paper presents a novel idea and the results are strong. My rebuttal comments were addressed well.

**Limitations:**

yes

**Paper Formatting Concerns:**

No major formatting concerns.

**Quality:**

3

**Strengths And Weaknesses:**

**Strengths**
1.  The paper tackles a highly relevant and open challenge: predicting conformational ensembles of biomolecules, which has broad applications in structural biology and drug design.
2.  The proposed method, ConforMix, is both novel and promising. Notably, it appears to outperform BioEmu—a state-of-the-art method trained on molecular dynamics data—in at least two respects: (i) recovery of native conformations under varying RMSD thresholds (Figures 2C and 2D), (ii) efficiency in free energy estimation (Section 4.1).
3.  The proposed approach requires no retraining and could be widely applicable to pretrained diffusion models.

**Weaknesses**

1.  Experimental details are missing, making it hard to fully interpret the results:
  - How were the proteins in Figures 2A and 2B selected? What exactly is being projected in the PCA plots?
  - Are the protein sets in Figures 2C and 2D the same as those used in the BioEmu paper? Please explain the metrics.
  - The conclusions from the RNA modeling experiment (starting line 230) are insufficiently supported. Are the grey structures in Figure 3B experimental references? Please support both the claim of lower performance on RNA than on proteins, as well as recapitulation of conformations from MDLu et al. 2025.
2.  Although Section 3.3 discusses free energy prediction and sample reweighting, these components do not appear to be used or evaluated in the experiments. In Section 4.1, free energy estimation seems to be performed using the BioEmu method rather than the proposed one. A comparison is needed.
3.  Runtime performance is not discussed.
4.  Some parts lack clarity:
  - TwistedSampling, central to ConforMix, is not described. A brief intuitive explanation in the main text and a detailed description in the appendix would be very helpful.
  - [Line 7] Mentions classifier guidance, but it does not appear in the methods—this is also mentioned on line 113.
  - [Line 149] It’s unclear what the variable j represents.
  - [Lines 280–283] The discussion is confusing. Is the proposed method actually faster than existing ones?
  - [Line 99] The conditioning variable is inconsistently denoted as both $input$ and $s$ later.
5.  Typos and minor errors
  - Line 78: “(,”
  - Line 83: “[ cite …”
  - Line 157: “[ CITE …”
  - Figure 2: Floating letter “B” in the figure

---

> ### Author Rebuttal · Authors · 2025-07-31
>
> We thank the reviewer for their enthusiastic evaluation of our method and for their helpful feedback. We are glad they found the approach to be "novel and promising." We have updated the text significantly based on the reviewer's comments and provide answers to their questions below.
>
> ## 1. Experimental details:
> ### a)
> **Selection of proteins Fig 2:** The proteins in Fig. 2A and 2B are meant to provide case studies of the behavior summarized in Fig. 2C and 2D, where Boltz-ConforMix captures experimental states and relevant motions for domain motion and cryptic pocket proteins. We selected examples that represent good multi-conformation sampling, which is seen across many but not all proteins in these sets. We have added similar visualizations for every tested protein to the Supplement so that a detailed comparison can be made.
>
> **PCA plots in Fig 2:** We have updated our text to better describe the principal component analysis and our interpretation of it. For these analyses, we transform each generated sample to internal coordinates by computing a distance matrix between all alpha-carbons, then perform a PCA across all Boltz-ConforMix generated samples for a given protein. To investigate what the principal components capture, we visualize samples at different positions along PC1. We also project experimental structures onto this PCA space to visualize where known conformations are positioned relative to our generated ensemble. For many, though not all, proteins, we find that Boltz-ConforMix produces samples along physically meaningful transition pathways that connect experimental conformational states (e.g., hinge open-to-close, domain in-to-out, transporter inward-to-outward facing). For instance, Fig. 2A/B show domain motions progressing from closed/hinge-inward to open/hinge-outward conformations through physically plausible intermediate states. These motions dominate the sampling of these systems. We now include PCAs for all proteins tested in the Supplement.
>
> ### b)
> Yes, the domain motion and cryptic pocket protein sets depicted in Figs 2C and 2D are the same datasets as in the BioEmu paper. The global and local RMSD metrics, respectively, are the ones used by BioEmu on these sets. Specifically, the global RMSD metric is the RMSD of the entire protein to the reference structures, which is used on the domain motion set. The local RMSD metric is the RMSD of a predefined region of the protein around the location of the cryptic pocket. We have updated the text to clarify these details and better reference their origin from BioEmu.
>
> ### c)
> Regarding the comparison of RNA modeling to molecular dynamics results, the grey structures depicted in Fig 3B are frames from MD simulations performed by Lu et al., which we use as reference data for this system since experimental structures do not exist for all of the states depicted. Thus, Fig 3B illustrates the extent to which Boltz-ConforMix sampling can recapitulate RNA conformational heterogeneity found in MD simulations. We have expanded our explanation of the reference structures accordingly. With regard to the performance on RNA modeling, we agree with the reviewer that the claim of lower performance is not sufficiently supported. We have not evaluated a larger dataset of RNA conformational variability and thus intended our remarks simply as a caution that performance may vary between proteins and other systems such as RNA. We have clarified the text accordingly.
>
> ## Free energy prediction and sample reweighting
> We believe that the quantitative components of our approach complement the qualitative understanding available from the conditional sampling alone. We have indeed evaluated these components with both BioEmu-ConforMix (described in the text) and Boltz-ConforMix (which we agree is not sufficiently covered in the text). With BioEmu, we compare the convergence of free energy estimates via our procedure and convergence of free energy estimates via direct unbiased sampling of BioEmu. Figure S4B illustrates a set of such convergence comparisons, showing that our procedure ("mbar", orange line) is more sample-efficient than unbiased estimation. We purposely selected these examples to be achievable via unbiased sampling. Consider a free energy difference of interest, $\Delta G_{a,b}$, between states $a$ and $b$. As $\Delta G_{a,b}$ becomes greater, convergence of the free energy estimate $\Delta \hat G_{a,b}$ derived from unbiased sampling becomes exponentially slower. Our approach of biased sampling followed by reweighting makes estimating such free energy differences substantially more practical.
>
> It is logical to examine the performance of our free energy estimation machinery with Boltz-ConforMix as well. A feasible amount of unbiased Boltz sampling will never sample much of the conformational heterogeniety we access with Boltz-ConforMix. This makes a quantitative comparison between our method and unbiased sampling almost trivial, since the (inaccurate) free energy estimate from unbiased sampling is simply $\infty$ for states not visited. Our approach enables more precise quantification of the Boltz distribution. We have revised our manuscript to describe these additional observations.
>
> ## Runtime performance
> The computational cost of generating $n$ samples using our default configuration of ConforMix+Boltz or ConforMix+BioEmu is approximately 3x that of generating $n$ samples from the underlying model via the standard SDE solvers. The twisted diffusion procedure involves two additional operations per denoising timestep compared to standard sampling: gradient-based guidance and particle resampling. There is some flexibility to tune these costs by scheduling these operations at only some steps of the denoising process, though we have not thoroughly explored the accuracy tradeoffs involved in doing so. However, we believe our results demonstrate that the benefits of exponentially improved rare-event sampling significantly outweigh additional sampling costs of this magnitude.
>
> ## Clarity
> We thank the reviewer for their notes on clarity and have updated the text to better explain the points they have identified.
> - TwistedSampling refers to the sequential Monte Carlo procedure of Twisted Diffusion Sampling [1]. We have included a description of this method in the revised text.
> - We have updated the text to describe the classifier guidance approach we leverage here.
> - The variable $j$ on line 149 indexes the set of conditional distributions we sample; these correspond to the bias potentials constructed from the set of RMSDs in line 4 of Algorithm 1.
> - Our method is overall substantially faster than unbiased sampling in cases where the states of interest are sufficiently different in free energy (one is rarely sampled), which is frequently the case in these models as well as in actual Boltzmann distributions. Please see also our remarks on runtime performance above.
> - The conditioning variable was indeed inconsistently denoted; we have corrected this in the text.
>
> ## Typos
> We thank the reviewer for identifying these typos and minor errors. We have resolved them.
>
> ## Questions
>
> ### Q1.
> We thank the reviewer for highlighting our pLDDT structure filtering approach. Their insightful question has prompted us to include a clearer explanation of and justification for this approach. ConforMix enables structure generation with arbitrary constraints, thus requiring a way to determine when the samples generated under these constraints (e.g. specific RMSD values from default predictions) are compatible with a protein's conformational flexibility. We initially tried filtering for physically feasible samples using average pLDDT over all residues, as the reviewer suggests. We found, however, that when we bias structure generation away from default predictions with RMSD values incompatible with a protein's physical constraints, the resulting samples often exhibit non-physical structure only in small protein segments. Since the majority of the predicted structure remains physically valid, the average pLDDT over all residues typically shows minimal decrease, making such a filter unreliable. In contrast, the pLDDT of these small non-physical segments decreases dramatically. We thus implemented a sliding window approach to ensure every sub-region of the protein maintains relatively high pLDDT/physical validity. We have updated the manuscript text to describe our rationale for choosing this metric. We also refer the reviewer to our response to Q2 from reviewer pToA for further details on the behavior of this filtering. Based on feedback from both reviewers, we have added a supplementary figure demonstrating filtering statistics across different proteins and RMSD ranges.
>
> ### Q2.
> The RMSD guarantees the reviewer asks about are an important point to clarify. As we note elsewhere, ConforMix can be used with arbitrary biasing potentials, but in the procedure outlined in Algorithm 1, we bias structure prediction $R$ Å away from the \textit{default conformation} predicted by Boltz/BioEmu/etc (considering only regions with secondary structure), rather than $R$ Å away from an experimental structure. In general, we find that generated samples satisfy their respective RMSD targets, although some of these samples contain physically implausible features and are removed by our filtering procedure. This process does not guarantee that generated samples will have a specific RMSD to experimental reference structures, however. The ability to recapitulate reference structures (e.g. open 16PK and closed 13PK for phosphoglycerate kinase) varies by protein and is summarized in Fig. 2C. We have added the best match RMSD values to each reference structure for all tested proteins to the Supplement.
>
> [1] Wu et al, "Practical and Asymptotically Exact Conditional Sampling in Diffusion Models", NeurIPS 2023.

---

> > ### Comment · Reviewer_61jC · 2025-08-04
> >
> > Thank you for the clarifications! I am keeping my score the same.

---

### Official Review · Reviewer_PcUq · 2025-06-23

**Clarity:** 3
**Significance:** 4
**Originality:** 3
**Rating:** 5
**Confidence:** 3

**Summary:**

In this paper, the authors propose ConforMix, a sampling method that employs Twisted Diffusion and Sequential Monte Carlo within a diffusion model. This method allows us to sample a wide variety of conformations, enabling us to capture the most important conformational changes for understanding biomolecular function.

**Questions:**

You might be planned as the next step, but will it yield results for predicting protein complexes and extending docking simulations?

**Ethical Concerns:**

["NO or VERY MINOR ethics concerns only"]

**Final Justification:**

ConforMix is an interesting study that proposes a method for sampling multiple conformations of important biomedical proteins. I increased my rating by one because I gained a better understanding during the peer review process, and I believe this manuscript will be useful for subsequent research.

**Limitations:**

Yes

**Quality:**

3

**Strengths And Weaknesses:**

Strengths:
This is an interesting method of extending a diffusion model-based generative model, which essentially predicts static three-dimensional structures, to dynamic three-dimensional structures that can be predicted without information on degrees of freedom. This method uses sampling with Twisted Diffusion and Sequential Monte Carlo and reweighting with the Multi-State Bennett Acceptance Ratio (MBAR) free energy estimation algorithm to guide and reweight conditioning generation. When incorporated into Boltz, it can predict structures closer to those revealed by experiments that Boltz alone cannot predict. This method is more sophisticated and scalable than the previously reported AFCluster, and its performance is comparable to that of BioEmu.

Weaknesses:
It seems that three-dimensional coordinate information is essential for input. The prediction accuracy is affected by the statistical uncertainty of MBAR, as mentioned in the limitations, and it may depend heavily on the accuracy of the extended diffusion model predictions. Therefore, validation experiments in a wet lab are ultimately required. However, this is not reason to lower the paper's rating, as this applies to all in silico models.

Minor point:
I found several typographical errors in the text. The following is a list of the ones we noticed:
Line 78: Excessive parentheses.
Lines 83 and 157: The reference appears to be incomplete.
Figure 3: "applicaation" is a typo for "application."
Line 235: “MDLu” is a typo for “MD Lu”.
Line 293: “hte” is a typo for “the”.

---

> ### Author Rebuttal · Authors · 2025-07-31
>
> We thank the reviewer for their positive evaluation of our method, observing that it allows us to sample a wide variety of conformations. We are glad they found the approach to be sophisticated and scalable. We appreciate their useful feedback and believe we address the limitations they describe in our responses below.
>
> The procedure we describe in Algorithm 1, which we used to produce the data visualized in Figures 2 and 3, does not require external 3D coordinate information to be supplied. Instead, an initial 3D conformation is generated at runtime from the diffusion model with default sampling (line 1) and this is then used as a reference state to construct biasing potentials (lines 2 and 3).  For instance, all results from Boltz-ConforMix shown in Figs 2, 3, and S1 are generated without any user-input 3D coordinates.
>
> Nevertheless, ConforMix readily accommodates alternative sampling objectives. For example, a user may provide a reference 3D conformation or other coordinates of interest (e.g. distances between regions of interest in a protein), and ConforMix can perform sampling along those degrees of freedom. E.g. samples in Fig S2 were generated using distance-based coordinates (biasing the diffusion model to generate samples with user-input distances between protein regions of interest). Based on the reviwer's comments, we have revised the manuscript to highlight the versatility of the method in accommodating objectives **with** or **without** input coordinates.
>
> We acknowledge that because our method runs at inference time, the quality of its predictions necessarily depends on the behavior of the underlying diffusion model, as well as the performance of Twisted Diffusion Sampling and MBAR. Despite these limitations, we believe our results demonstrate the utility of ConforMix. We also acknowledge that wet lab validation experiments for computational results play an essential role in the scientific process. We hope that our tool will be useful to wet lab scientists in guiding their experimental investigations.
>
> We thank the reviewer for their careful reporting of typographical errors, which we address in the revised manuscript.

---

> > ### Comment · Reviewer_PcUq · 2025-08-07
> >
> > I greatly appreciate how attentive the authors were to my concerns and questions. I apologize for misinterpreting the text regarding the input data required by the model. May I confirm that the only input data required by the model is the primary sequence of amino acids, as per Boltz and AlphaFold? Can post-translational modifications of proteins be inferred to some extent, e.g., the regulation of conformation by phosphorylation and glycosylation?

---

> ### Author Response · Authors · 2025-08-08
>
> Thanks for your kind response!
> - We confirm that the only input data needed to run ConforMix with RMSD-based biasing (as was done for results in Figs 2, 3, S1) is the primary sequence of amino acids. ConforMix can *alternatively* be used with other types of biasing where users supply descriptors of interest to vary (e.g. distance between specified regions in a protein as in Fig S2).
> - If the underlying model (e.g. AlphaFold, Boltz) handles post-translational modifications, then ConforMix could be applied as you suggest. However, we have not evaluated our approach for PTM-induced conformational changes, so we cannot currently assess whether Boltz-ConforMix would accurately predict PTM effects on protein structure.

---

> > ### Comment · Reviewer_PcUq · 2025-08-08
> >
> > Thank you very much. My questions have mostly been answered.

---

### Official Review · Reviewer_pToA · 2025-07-01

**Clarity:** 3
**Significance:** 3
**Originality:** 3
**Rating:** 5
**Confidence:** 3

**Summary:**

The authors developed ConforMix, an inference-time scaler for diffusion-based structure predictor of biomolecular systems, to enable multi-state sampling.
Using pseudo-potentials under different conditions and Twisted Diffusion Sampling (TDS), they were able to efficiently sample an extensive conformational space.
Based on these results, the authors connected their method with the long-standing MBAR (multistate Bennett acceptance ratio) method, which is a free energy estimation method based on system perturbation. This allows for faster convergence of MBAR's energy estimation.
The authors chose important and quite different biomolecular systems to demonstrate ConforMix's multi-state sampling efficiency and adopted benchmark datasets to show its efficiency in free energy estimation.

**Questions:**

# Questions
1. I suggest using supplementary data 1 of [1], which consists of 92 fold-switching protein conformations. In [1], the authors tested the capability of AF-based models to sample multiple conformations of fold-switch proteins and measured the success rate of sampling multi-state conformations. Alternatively, the dataset used in EigenFold (originally constructed from [3]) can be used for benchmarking. This benchmark implicitly evaluates the models' ability to sample multiple states using TM-score and residue-wise flexibility as metrics. Conducting such benchmarks would improve the paper's impact.
2. Boltz only trained to reproduce the static (mostly stable) PDB structures, not the physical energy function $-\nabla U(x)$. Thus, the drift that Boltz learned is not a physical force. It seems that using a pseudo-potential like RMSD might lead to highly physically invalid structures. I think that the authors' choice to use a pLDDT value higher than 20% for filtering generated structures implies that most of generated samples are in a physically invalid (off-manifold) state. Could the authors provide the statistics on physical validity of ConforMix-generated samples compared to Vanilla-Boltz-generated samples?
3. The accuracy of MBAR estimation depends on the number of samples from each state and the overlap between states. To me, efficient sampling of various states and MBAR accuracy seem to be mutually exclusive. Is there any potential bias triggered by a small number of samples due to efficient exploration?
4. The dynamic state visualization results of ConforMix presented in the main text and supplementary data are very impressive. By what criteria were these examples selected among the numerous structural ensembles generated?
5. I think multi-state conformation sampling is more important for novel proteins rather than the systems that we know about.
Is this right?
If so, how can one use ConforMix to fully novel protein, where no information about multiple conformational states are available?
User may select the suitable pseudo-potential with appropriately consider physical validity, and then should properly select the generated conformations.
If the authors provide proper ways to design pseudo-potential and conformer selection criteria (which is related to **W2** and **Q4**), this would enlarge the impact of the work.

# Typos
1. In Figure 3 caption's 4'th line: applicaation -> application
2. In line 276: an series -> a series
3. In line 293: What does mean by hte?
4. The year is missing from reference 4.

[1] Chakravarty, Devlina, et al. "AlphaFold predictions of fold-switched conformations are driven by structure memorization." Nature communications 15.1 (2024): 7296.
[2] Jing, Bowen, et al. "Eigenfold: Generative protein structure prediction with diffusion models." arXiv preprint arXiv:2304.02198 (2023).
[3] Chakravarty, Devlina, and Lauren L. Porter. "AlphaFold2 fails to predict protein fold switching." Protein Science 31.6 (2022): e4353.

**Ethical Concerns:**

["NO or VERY MINOR ethics concerns only"]

**Final Justification:**

My previous questions were considered well. In particular, the addition of new benchmark results, the conformation selection strategy in PCA, the detailed explanation of the trade-off in MBAR, the applicability to novel proteins (as well as characterized proteins) and its importance, were helpful. The authors also provided additional empirical results about physically valid structures and their reasoning behind filtration schemes to obtain them.

**Limitations:**

yes

**Paper Formatting Concerns:**

- Supplementary figure S4 and the code are showing partially as they pass the width limit.

**Quality:**

2

**Strengths And Weaknesses:**

# Strengths
1. The proposed method uses system-dependent conditions with potentials for TDS and has demonstrated its ability to transform protein structure prediction, such as Boltzmann machines trained with static conformations, into multi-state conformational samplers.
2. Fast free energy estimation, based on a combination of MBAR, seems to have the potential to be used in the analysis of protein dynamical systems.
3. The motivation of the paper and the method and the experimental results are well aligned.
The paper's core result is that Boltz-ConforMix performs well in multi-state sampling, demonstrating its performance on domain and cryptic pocket motion datasets.
Two systems, which used for case studies, are quite different (SemiSWEET sugar transporter is a relatively less dynamic protein system where the hammerhead ribozyme is a more dynamic RNA system), demonstrating the proposed method's effectiveness.

# Weakness
1. **Lack of validations for generated conformations**. [related to **Q2**]
Boltz-ConforMix in Figure 2(c) shows that it beats the BioEmu's performance under the range around 3Å.
How can it be possible only using RMSD compared to the initially generated conformation from Boltz?
Given the fact that (a) BioEmu learned vast conformational space with dynamical data (b) Boltz only learned mostly stable data, I think it is still hard with ConforMix to generate physically validated structures while covering more extensive conformational space than BioEmu.
The authors should provide an analysis about the structural validity of Boltz-ConforMix.

2. **Potential biased selection of structural visualzation**. [related to **Q4**]
How did author selected the visualization results from data?
Since several results are picked from low-density region of the PCA results, it seems that the authors chose the results with specific criteria, which should be mentioned in the manuscript.
For example, were structures chosen based on their distance from a cluster center, their energy, a specific structural motif, or another quantitative measure?

3. **Incomplete manuscript**.
Although the final quality of the paper does not depend on its initial completeness, most papers submitted to the conference are well-formatted, with complete citations and figures.
However, this paper has several critical issues.
- Incomplete citations (lines 83 and 157)
- Line 157 reveals the authors' names, which violates the anonymity requirement
- In Figure 2(b), the label under *hinge outward* structure seems to be fixed (0.46Å to 1DV2) -> (?Å to 1DV1)
- In supplementary figures S3(c) and S3(d), it seems that the axes are swapped, making it difficult to understand whether BioEmu and ConforMix generated similar conformational distributions

4. **Understanding the manuscript is quite hard due to the lack of information**.
- RMSD metrics used for each domain motion and cryptic pocket sets seems to be global and local RMSDs (inferred from BioEmu). These information is critical for evaluating the results of the paper so the authors should give more information about each results' metrics.
- In the algorithm section, RMSD first appears, but there is no explanation of why the mask is generated from the rigid element or what the mask does in RMSD. (A detailed explanation is provided in Section 4.1, especially in line 274.) Using captions for detailed explanations would help readers better understand.
- The authors should explicitly inform the readers about related supplementary section. (e.g. supplement -> supplementary figure 4)

---

> ### Author Rebuttal · Authors · 2025-07-31
>
> We thank the reviewer for their thorough reading and very constructive suggestions. We are glad they observe the potential of our method to predict protein dynamics. Based on their comments, we have significantly updated our manuscript and run further benchmarking metrics and comparisons. We hope that our responses below address their concerns.
>
> ### 1. Validating generated conformations
> This is an excellent question. We believe our findings add to the growing literature (primarily based on MSA perturbations) that shows that even models trained for single-structure prediction can generate varied conformations.
>
> Although BioEmu is finetuned on MD simulations and experimental stability data, it is initially pretrained on only clustered PDB data. This pretrained  model still performs remarkably well at conformational sampling (see e.g. BioEmu Fig 2, S1, S2, S4). In fact, for many proteins, the finetuning does not appear to significantly improve the conformational coverage. Thus, there is precedent for diffusion-based sampling of conformational changes given only training on static structures.
>
> We appreciate the reviewer's suggestion to analyze structural validity of these samples. We now include statistics on outliers in Ramachandran angles, backbone bond angles, and bond lengths in the manuscript. These results show slightly increased outliers for Boltz-ConforMix samples compared to vanilla Boltz samples. We summarize results on the domain motion protein set below. A bond length or bond angle is an outlier if its value is beyond $4 \sigma$ of the reference value. A residue, excluding Gly and Pro, is a Ramachandran outlier if its $\phi/\psi$ angles lie outside standard regions. Each table entry gives the outlier rate (number of outliers divided by total number of backbone bonds, backbone bond angles, or residues), averaged across all samples collected of all proteins.  These results show that ConforMix samples are generally physically valid.
>
> | Fraction of outliers in: | Boltz | Boltz-ConforMix no filtering | Boltz-ConforMix w/ filtering |
> |-|-|-|-|
> | Rama. angles |  $0.0088 \pm 0.0055 $ | $0.0142 \pm 0.0047$ | $0.0124 \pm 0.0054$|
> | Backbone bond angles | $(2.2 \pm 4.2) \times 10^{-5}$ | $(31.4 \pm 32.1) \times 10^{-5}$ | $(8.2 \pm 4.7) \times 10^{-5}$|
> | Backbone bond distances |  $(2.3 \pm 1.8) \times 10^{-5}$ | $(28.2 \pm 21.3) \times 10^{-5}$ | $(14.1 \pm 8.7) \times 10^{-5}$|
>
>
> ### 2. Selection of visualized structures
>
> This is a good question. For each protein sampled with Boltz-ConforMix, we perform a principal component analysis on the generated samples. We chose samples to visualize with the primary goal of illustrating what motion is captured by the largest PCs. Please see our response 1a to reviewer 61jC for more details.
>
> To address the reviewer's concern and better demonstrate how ConforMix samples physically-relevant transitions, we have:
> - Expanded our description of the PCA procedure (including how visualized samples were selected) and what these analyses indicate about transition paths between experimental structures.
> - Provided PCAs for all proteins in the supplement.
> - Included analyses of the "trajectories" generated by Boltz-ConforMix samples. E.g. projection analyses quantifying on- vs off-pathway conformational variation between the experimental states and fill ratio analyses (quantifies how well samples map onto the path between known states, from [1]).
>
> Please also see our response to Q4.
>
> ### 3 and 4. Corrections
> We thank the reviewer for pointing out errors.
> - We have corrected the citations, figure axes, and Fig. 2B labels. We also reference supplementary figures in the text.
> - We apologize for the inadvertent internal comment on line 157. However, we wish to note that this comment referred to citing material from an external source and does not reveal the name of any authors of this work.
>
> - The reviewer is correct that we use the metrics from BioEmu (global RMSDs for domain motion and local RMSDs for cryptic pocket sets). We have updated the text to reference this.
>
> - The question about the choice of residue mask for computing RMSDs is an excellent one. Most proteins contain both flexible loop regions and rigid secondary structure elements. During initial testing we computed RMSD on all residues, which revealed that models preferentially sampled loop motion. Though these are usually physically valid, loop flexibility is typically easy to predict via other means, so we aimed to concentrate sampling efforts on larger changes that rely on rigid element motion. We thus compute RMSD only on residues within rigid secondary structure elements (helices and sheets). Forcing RMSD changes measured on these regions guides Boltz-ConforMix to recapitulate larger dynamics such as domain motions and transporter cycling. This mask does not limit loop regions from moving during diffusion; indeed, loops often must move to allow rearrangements of rigid elements. We updated the text to better describe this idea.
>
> ## Questions
>
> Q1. We agree that further metrics and benchmarks would improve our paper. Following the reviewer's recommendation, we updated our analyses and manuscript to include TM-score analyses. E.g. for domain motion set:
>
> |Method|Default Boltz|Boltz-AFCluster|Boltz-Conformix|
> |-|-|-|-|
> |TM-score of best models (mean$\pm$STD)|$0.83\pm0.13$|$0.84\pm0.14$|$0.88\pm0.13$|
>
> We are also exploring residue-wise flexibility metrics.
>
> The recommendation to test the fold-switching benchmark is a good one. We are running these proteins and will include the results in the final version of the manuscript. In the interim, we have also run the open-closed (OC23) dataset from [1], excluding six proteins that overlap with the domain motion set. Please see our response to reviewer YdFU for statistics on this dataset.
>
> We are also benchmarking against more recent AF-based models including AFsample2 [1] and CF-Random [2].
>
> Q2. The reviewer identifies a key design consideration in our filtering approach. Our conditional diffusion framework allows generation under arbitrary constraints, but some constraints would require a protein to assume a physically implausible conformation. Since we do not always know a priori if a given constraint is reasonable, our goal in filtering is to screen out samples that violate physical norms. The 10-residue sliding window pLDDT approach serves as our mechanism for identifying when biasing constraints exceed a protein's natural flexibility limits. We expect that proteins that undergo large motions relative to the initial structure will remain physically plausible at greater RMSD values, while proteins with limited motion will show validity dropoffs at lower RMSD values. Based on the reviewer's question, we analyzed filtering statistics across our datasets and confirm this trend. The fraction of filtered structures increases with the biasing RMSD value, e.g. for the domain motion set:
>
> |RMSD|Filter Ratio|
> |-|-|
> |0|$0.0\pm0.0$|
> |6|$0.02\pm0.06$|
> |10|$0.18\pm0.36$|
> |18|$0.36\pm0.44$|
>
> These statistics vary substantially between proteins based on their degree of conformational flexibility. For example, P06766 (10.3 Å experimental RMSD) accepts 97\% of structures up to 12 Å, then drops to zero acceptance. P02911 (5.0 Å experimental RMSD) accepts 98\% under 6 Å, with no acceptance above 9 Å. RMSD acceptance thresholds don't always match experimental ranges exactly, Boltz-ConforMix samples conformations beyond experimentally observed states while often remaining physically plausible. For example, biotin carboxylase (Fig 2B) samples states more extended than the known "outward hinge" conformation. We have added a figure showing filtering statistics across proteins and RMSD ranges. W also provide physical validity statistics (see W1).
>
> Q3. MBAR provides unbiased estimators of the partition functions. As the reviewer notes, the variance of those estimates depends on the number of samples and the overlap between states. Please see our response to reviewer YdFU Q1 for further discussion.
>
> Q4. Fig 2: these are typical cases where ConforMix captures experimental states and motion, a pattern seen across many proteins (see coverage plots) but not all. Our supplement now shows visualizations for each tested protein. We also updated Fig 2 to include one domain motion and one cryptic pocket representative.
> Transporters and RNA (Fig 3, S1): we consulted with collaborators on proteins with physically relevant motion. We focused on these transporters as they have >1 chain (not accessible with most previous methods) and exhibit >2 relevant states, allowing us to test whether ConforMix can recapitulate the full set. The ribozyme was chosen as MD data was available to validate our generated  states.
>
> Q5. The reviewer raises an excellent point about novel protein applications. Our RMSD-based approach can be applied to any protein sequence without prior knowledge: it uses only the structure prediction model's default output to systematically bias sampling at progressive RMSD distances, and relies solely on model-derived confidence metrics for filtering.  While ConforMix supports different potentials (see our response to PcUq), this RMSD approach (used for Fig 2-3) is applicable without information about multiple conformational states. For any protein, users can: 1) predict a default structure, 2) sample with bias potentials at increasing RMSD distances (e.g. 2-20 Å used here), and 3) filter for plausible conformations.
>
> We note that conformational sampling is valuable not only for novel proteins but also for characterized proteins where dynamics remain experimentally challenging to capture (e.g., transporter mechanisms, binding transitions).
>
> [1] Kalakoti & Wallner. "AFsample2 predicts multiple conformations and ensembles with AlphaFold2." Comm Bio (2025).
> [2] Lee et al. "Large-scale predictions of alternative protein conformations by AlphaFold2-based sequence association." Nat Comm (2025).

---

> > ### Comment · Reviewer_pToA · 2025-08-03
> >
> > Thanks to the authors for their detailed responses. My previous questions were considered well. In particular, the addition of new benchmark results, the conformation selection strategy in PCA, the detailed explanation of the trade-off in MBAR, the applicability to novel proteins (as well as characterized proteins) and its importance, were helpful. The authors also provided additional empirical results about physically valid structures and their reasoning behind filtration schemes to obtain them.
> >
> > **Therefore, I'll increase my rating.**
> >
> > Just out of curiosity, I have a question about how the ConforMix works.
> > I'm surprised that physically valid structures can be generated with only an RMSD-based pseudo potential. I'm curious about the author's insight into how ConforMix generates physically valid structures.
> > In my opinion, Vanilla Boltz already learned the drift term to make physically valid structures, which means that physically valid structures are placed in their low-energy (high-probability) potential (not physical energy) wells. The local minimum structure for each potential well represent different conformation (in terms of RMSD). The core role of ConforMix in inference phase is simply renormalize the conformational space's probability considering additional RSMD term.
> >
> > Here's my question: If there are only two conformations of cyclohexane---the boat and chair forms---and the structure prediction model has only learned the boat form (i.e., overfitted to generate the boat form conformation), then can ConforMix still generate physically valid chair-form structures with RMSD pseudo-potential? Why or why not?

---

> > > ### Author Response · Authors · 2025-08-06
> > >
> > > We thank the reviewer for their positive opinion and their questions. As you mention, the additional biasing term (e.g. RMSD term) in ConforMix alters the probability distribution sampled by the diffusion model, resulting in exploration of alternate states. Generating relevant alternate conformations depends on the underlying model capturing some molecular physics, even if it is imperfect.
> > >
> > > The ConforMix strategy using RMSD potentials relies on accessing conformations that already exist within the latent space of the model. For a toy model of cyclohexane, if the model hasn't captured anything about conformations other than the boat form (e.g. any perturbations away from the boat are equally unlikely), we’d expect biasing away from this structure to result in random perturbations, not a specific transition to a chair-form. However, if the model captured some information about chair flips (be it from cyclohexane or other molecules that undergo related structural transitions), ConforMix may be able to overcome the model’s boat bias to recapitulate the chair conformation. We note that since Boltz generates samples in $\mathbb R^{n \times 3},$ all conformations can in principle be generated.
> > >
> > > The perhaps surprising generation of physically valid structure with only RMSD-based potentials with ConforMix lead us to some relevant takeaways:
> > > - We believe that protein structure diffusion models such as Boltz have learned "incomplete physics". That is, they have learned relevant motions and physical constraints in some but not all cases. The strategy of using ConforMix with RMSD potentials has shown that enough information has been learned to enable sampling additional conformations for many proteins, but not to perfectly reconstruct conformational space.
> > > - More broadly, we believe our enhanced sampling approach will continue to be useful as underlying model accuracy improves.

---

### Official Review · Reviewer_YdFU · 2025-07-02

**Clarity:** 3
**Significance:** 3
**Originality:** 2
**Rating:** 5
**Confidence:** 3

**Summary:**

The paper proposes the algorithm ConforMix for sampling from a distribution of conformations. ConforMix requires a pretrained generative model as its backbone and aims to improve the prediction of conformations produced by this model. ConforMix combines three separate algortihmic components:
- To reweight samples, the partition function of the imposed condition is estimated via the multistate Bennett acceptance ratio.
- Twisted Diffusion is applied to condition the sampling process on any differentiable feature.
- The generated samples of ConforMix are rejected if any 10-residue sliding window exhibits an average pLDDT value more than 20% lower than that of the default prediction.

ConforMix is implemented in Boltz to generate diverse conformations rather than a single conformation for a given input, and in BioEmu to improve free energy estimation. The authors commit to releasing these implementations upon publication, and code snippets are already provided in the supplementary material.

**Questions:**

- What is the computational cost of ConforMix? How long is the sampling time of ConforMix+Boltz compared to sampling with Boltz only? How long is the sampling time of ConforMix+BioEmu compared to BioEmu only?
- Is it necessary to estimate the normalizing constants  $Z_{1},...,Z_{J}$ appearing in line 173, as well as in Eq. (3) and Eq. (4), in order to solve the linear system described in lines 178–181? If so, how do the authors estimate these normalizing constants?
- How large is $J$ in the experiments of the paper? Is it clear that the system of $J+1$ unknowns admits a (unique) solution?

### Minor comments:
- To open up the content of the paper to people with a computer science background only, it would help to add references to the first paragraph of the introduction. The second paragraph of the introduction is missing in the reviewers opinion some references as well.
- there are also no references in the second paragraph of the introduction
- (Line 83): internal comment
- (line 92): broken reference
- (line 151): bracket and period are interchanged.
- (Line 157): internal comment
- (line 235): is "MDLu et al." correct?
- (line 292): "the" instead of "hte"

**Ethical Concerns:**

["NO or VERY MINOR ethics concerns only"]

**Final Justification:**

The reviewer considers this a valuable contribution that advances conformational sampling by broadening the set of accessible protein states.

**Limitations:**

The authors have sufficiently addressed the limitations of their work.

**Paper Formatting Concerns:**

The reviewer has no concerns regarding the formatting of the paper.

**Quality:**

3

**Strengths And Weaknesses:**

### Strengths
- ConforMix is implemented in the open-source tools Boltz (an open-source, diffusion-based structure prediction model) and BioEmu (a diffusion-based model for conformational sampling). The authors commit to releasing these implementations upon publication, enabling future research to easily test ConforMix and integrate it with their own conformation-generating models.
- ConforMix empirically broadens the accessible states of proteins, as demonstrated in four protein studies using Boltz. When combined with BioEmu, ConforMix can generate ensembles that extend beyond those sampled with BioEmu alone. In addition, applying ConforMix to RNA structures shows that it can recapitulate transitions observed in molecular dynamics simulations.
- The advantage of integrating ConforMix is demonstrated for both a static conformation-generating model (Boltz) and a dynamic one (BioEmu).

### Weaknesses
- The paper does not introduce new methodologies but rather combines existing methods into an ensemble tailored to improve the prediction of conformations generated by a given model, whether static or dynamic. Nonetheless, the reviewer acknowledges the careful application of these methods, which leads to practical improvements.

---

> ### Author Rebuttal · Authors · 2025-07-31
>
> We thank the reviewer for their careful reading and helpful feedback, and are glad they find that ConforMix demonstrates significant sampling advantages and recapitulates molecular transitions. While our method stands on prior algorithmic work, we believe that applying this combination of methods to protein dynamics is innovative and nontrivial.
>
> Although we highlight individual case studies in the paper, we note that we demonstrate application of this method to the domain motion and cryptic pocket protein benchmark sets from BioEmu, which contain 22 and 34 proteins respectively; statistics across these sets are shown in Fig 2.
>
> Based on reviewer comments, we have included or plan to include benchmarking updates including the following:
>
> - We have included additional TM-score analyses of method performance. E.g. for domain motion set:
>
> |Method|Default Boltz|Boltz-AFCluster|Boltz-Conformix|
> |---|---|---|---|
> |TM-score of best models (mean$\pm$STD)|$0.83\pm0.13$|$0.84\pm0.14$|$0.88\pm0.13$|
>
> - We have updated our Supplementary Material to include detailed RMSD and PCA analyses (similar to those in Fig 2A and 2B) for all proteins.
>
> - We included benchmarking on the open-closed (OC23) dataset from [1]. Summary statistics comparing Boltz-AFCluster and Boltz-ConforMix are as follows:
> |RMSD Threshold (Å)|1|2|3|4|5|
> |-|-|-|-|-|-|
> |Boltz-AFCluster|0.3|0.77|0.83|0.90|0.97|
> |Boltz-ConforMix|0.39|0.81|0.99|1.0|1.0|
>
> |Method|Boltz-AFCluster|Boltz-Conformix|
> |-|-|-|
> |TM-score of best open state models (mean$\pm$STD)|$0.82\pm0.16$|$0.84\pm0.17$|
> |TM-score of best closed state models (mean$\pm$STD)|$0.85\pm0.15$|$0.87\pm0.16$|
>
> - We will add benchmarking on a set of 92 fold-switching proteins, as suggested by Reviewer pToA.
>
> - We will add additional benchmarking against more recent AF-based models including AFsample2 [1] and CF-Random [2].
>
> We believe that we demonstrate the wide applicability of our approach by benchmarking on various datasets as well as case studies (see Fig 3).
>
> ### Responses to questions
>
> We hope that our below responses to the reviewer's questions will provide additional information about sampling and estimation.
>
> Q1. The computational cost of generating $n$ samples using our default configuration of ConforMix+Boltz or ConforMix+BioEmu is approximately 3x that of generating $n$ samples from the underlying model via the standard SDE solvers. The twisted diffusion procedure involves two additional operations per denoising timestep compared to standard sampling: gradient-based guidance and particle resampling. There is some flexibility to tune these costs by scheduling these operations at only some steps of the denoising process, though we have not thoroughly explored the accuracy tradeoffs involved in doing so.
>
> The key question is whether this sampling is ultimately less costly than the alternative: direct estimation from unbiased samples from the underlying model (e.g. Boltz, BioEmu). This depends on the quantity one is trying to estimate. For example, consider free energy estimation. In regimes where conformational states have modest energy differences (1-2 kcal/mol at 300 K), direct unbiased sampling may be sufficient: the higher-energy state will be sampled with probabilities of 1/5 to 1/30. However, important conformational changes often have associated energy differences of  >5 kcal/mol at 300K, resulting in a sampling probability of \~1/4500 for the higher-energy state. As the free energy difference increases, unbiased sampling becomes exponentially less efficient at estimating the free energies, and so the information provided by guided sampling is of critical value. We believe our results demonstrate that the benefits of exponentially improved rare-event sampling significantly outweigh additional sampling costs of this order.
>
> Q2. For simple choices of the twisting potential, $Z_j$ can often be calculated analytically. For example, if $g_j$ in the harmonic potential described on line 172 is one-dimensional, then the twisting potential has the normalization constant $\sqrt{\pi/k}.$ However, it is actually not necessary to estimate $Z_j$ in order to solve the linear system. The unknowns that MBAR estimates from the Eqn (4) system are the partition functions of the conditional distributions we are actually sampling, i.e. the quantities $\bar Z_j := p(y_j) Z_j$. These combined partition functions are sufficient for our purposes, because they enable us to compute importance sampling weights $w_i$ for all samples in the unbiased distribution $p(x),$ as suggested by Eqn (1). Using the reconstructed unbiased distribution (but with additional information about rare regions, thanks to the conditional sampling), one can then estimate arbitrary observables, including $p(x_j)$.
>
> Q3. For our purposes $J$ (the number of conditional distributions sampled) is typically in the range of 10-50, although larger $J$ can be practically accommodated if necessary. The system of equations has a unique solution, up to a constant factor in all of the estimated partition functions (see [3] and [4]). The practical implication of this free parameter is simply that we are not estimating the absolute partition functions of each state, but rather relative partition functions. Thus, we can use the partition functions as ratios of the form $\bar Z_a/\bar Z_b$, which is precisely the case when calculating importance sampling weights.
>
> We also thank the reviewer for their minor comments. We have updated the manuscript to provide additional references in the introduction and correct the other errors that the reviewer identified.
>
> [1] Kalakoti, Yogesh, and Björn Wallner. "AFsample2 predicts multiple conformations and ensembles with AlphaFold2." Communications Biology 8.1 (2025): 373.
> [2] Lee, Myeongsang, et al. "Large-scale predictions of alternative protein conformations by AlphaFold2-based sequence association." Nature Communications 16.1 (2025): 5622.
> [3] Shirts, Michael R., and John D. Chodera. "Statistically optimal analysis of samples from multiple equilibrium states." The Journal of chemical physics 129.12 (2008).
> [4] Tan, Zhiqiang. "On a likelihood approach for Monte Carlo integration." Journal of the American Statistical Association 99.468 (2004): 1027-1036.

---

> > ### Comment · Reviewer_YdFU · 2025-08-02
> >
> > The reviewer appreciates the detailed response, clarifications, and additional quantitative evaluation. I will maintain my score for now and will reflect further and discuss with the other reviewers before making a final decision.

---

### Note · Authors · 2025-08-12

We thank our reviewers again for their helpful comments and positive opinions of our paper. Our method, ConforMix, is an inference-time algorithm to enable sampling and quantitative free energy estimation of biomolecular conformations that are difficult or impossible to access via unbiased sampling. As the reviewers comment, "ConforMix tackles a highly relevant and open challenge," "empirically broadens the accessible states of proteins" and is "more sophisticated and scalable than AFCluster." We believe that our results demonstrate both the utility of ConforMix in the current generation of models (Boltz and BioEmu) and its applicability to future models that more closely approximate the Boltzmann distribution.

---

### Decision · Program_Chairs · 2025-09-17

**Decision:**

Accept (spotlight)

**Comment:**

This paper introduces ConforMix, an inference-time sampling framework for biomolecular diffusion models. It integrates (a) twisted diffusion sampling, (b) pseudo-potentials for conformation landscape exploration, and (c) MBAR-based sample reweighting for free energy estimation. The method is implemented for Boltz and BioEmu and verified on proteins and RNAs.

The paper is quite strong in a sense that it points out a novel research direction with high practical relevance. I also find the experiments on BioEmu and Boltz to be appealing for practitioners.

During the rebuttal period, reviewers raised concerns about (a) computational cost, (b) unclear details on MBAR, (c) Boltz training on static structures, and (d) frequent typos. A reviewer also suggested additional benchmarks like fold-switching proteins and OC23 dataset. The authors have resolved this concerns in a satisfiable way, and all the reviewers expressed excitement about this paper.

Overall, I believe this is a strong paper and recommend acceptance.